# Attitudes toward Homosexuality among Nurses in Taiwan: Effects of Survey Year and Sociodemographic Characteristics

**DOI:** 10.3390/ijerph18073465

**Published:** 2021-03-26

**Authors:** Huang-Chi Lin, Yi-Chun Lin, Yu-Ping Chang, Wei-Hsin Lu, Cheng-Fang Yen

**Affiliations:** 1Department of Psychiatry, School of Medicine, College of Medicine, Kaohsiung Medical University, Kaohsiung 80708, Taiwan; cochigi@kmu.edu.tw; 2Department of Psychiatry, Kaohsiung Medical University Hospital, Kaohsiung 80708, Taiwan; 3Department of Nursing, Kaohsiung Medical University Hospital, Kaohsiung 80708, Taiwan; pharaohnile@gmail.com; 4School of Nursing, The State University of New York, University at Buffalo, New York, NY 14214-3079, USA; yc73@buffalo.edu; 5Department of Psychiatry, Ditmanson Medical Foundation Chia-Yi Christian Hospital, Chia-Yi City 60002, Taiwan

**Keywords:** attitude, gay, homosexuality, lesbian, nurses

## Abstract

This study aimed to compare the level of attitudes toward homosexuality among nurses in Taiwan between 2005 and 2017 and with various demographic characteristics, as well as the roles of demographic characteristics in the changing trend of attitudes toward homosexuality between 2005 and 2017. This survey study recruited nurses from three hospitals in 2005 (Survey 2005, *N* = 1176) and 2017 (Survey 2017, *N* = 1519). Participants’ four dimensions of attitudes toward homosexuality, including condemnation, immorality, avoiding contact, and stereotypes, were assessed using the Taiwanese version of the Attitudes Toward Homosexuality Questionnaire. The results demonstrated that nurses in 2017 exhibited lower levels of avoiding contact with lesbian and gay patients and stereotypes toward homosexuality but higher levels of condemnation of gay and lesbian individuals and perceptions of gay and lesbian individuals as immoral than did nurses in 2005. Age moderated changes in some dimensions of attitudes toward homosexuality from 2005 to 2017. The need to develop training programs aimed at improving not only the quality of nursing skills but also their negative attitudes regarding homosexuality is urgent.

## 1. Introduction

### 1.1. Health Needs of Sexual Minority Individuals

Lesbian and gay individuals continue to constitute a vulnerable and marginalized group in healthcare. Lesbian and gay individuals have many healthcare needs, including the management of chronic diseases, mental health issues, interpersonal violence, and preventive care for risky behaviors [1]. According to the minority stress hypothesis, heterosexism-based stigma toward homosexuality results in chronic stress and mental health problems among lesbian and gay individuals [2]. Therefore, Healthy People 2020 identified increasing access to quality healthcare for lesbian and gay individuals as a priority for further research and intervention [3]. According to a comprehensive report published by the Institute of Medicine [4], increased attention should be paid to the health issues of lesbian and gay individuals.

### 1.2. Efforts to Integrate Gender into Nursing and Medical Education

Taiwan has made significant progress in gender equality in the past two decades. Taiwan ratified the Convention on the Elimination of all Forms of Discrimination Against Women (CEDAW) in 2007 to elevate the standard of gender rights in the country and advance gender equality; the government also promulgated the Enforcement Act of CEDAW in 2012 to make CEDAW provisions effective as domestic law [5]. The government also promulgated the Gender Equality Policy Guidelines in 2011 to establish the directions of administrative politics of gender equality [6]. Moreover, the government formulated the Gender Equality Education Act in 2015 to eliminate gender discrimination, safeguard human dignity, and establish education resources and environments that epitomize gender equality [7]. The rights of sexual minority individuals also received attention in Taiwanese society.

Regarding healthcare, the White Paper on Taiwanese Medical Education in 2002 firstly revealed the importance of integrating the concept of gender equality into medical education [8]. The Ministry of Health and Welfare in Taiwan amended the Healthcare Professionals Registration and Continuing Education Guidelines in 2016 to incorporate gender topics as compulsory credits for all healthcare professionals [9]. However, compared with the health needs of women, the health needs of sexual minority individuals have been less emphasized in Taiwan. Given that nurses are important healthcare providers, several professionals called for enhancing nurses’ competencies of healthcare for sexual minority individuals [10,11,12,13].

### 1.3. Attitudes toward Homosexuality among Healthcare Providers

A review study found that lesbian and gay individuals tend to feel excluded from healthcare services because of judgmental and heteronormative attitudes, practitioners’ ignorance of sexual minority issues, lack of sexual minority knowledge, and lack of respect and sensitivity [14]. Discrimination and sexual stigmatization by healthcare providers may force lesbian and gay individuals to delay seeking care; they may conceal their sexual orientation and refuse to reveal it to healthcare providers [1,15,16,17]. Therefore, evaluating healthcare providers’ attitudes toward homosexuality is crucial in the development of training programs by educational and clinical professionals to overcome prejudice, stereotypes, stigma, and silence on lesbian and gay issues among healthcare providers.

### 1.4. Attitudes toward Homosexuality among Nurses

Some nurses are underinformed regarding lesbian and gay issues, partially because of general heterosexism and conservatism [18,19]. Heterosexism and conservatism may increase the level of sexual prejudice among healthcare providers and prevent them from receiving the comprehensive training needed to develop a culturally competent ability in caring for lesbian and gay patients [20]. Both a lack of training and prejudice in healthcare providers may result in poor patient–provider communication and compromise patients’ adherence to and satisfaction with the care they receive [21]. Nurses in all areas of practice encounter lesbian and gay patients; thus, they should be attuned to the needs of these individuals [19].

### 1.5. Factors Related to Attitudes toward Homosexuality among Healthcare Providers

In 2005, a study conducted in Taiwan surveyed the attitudes of nurses toward gay and lesbian individuals and the reasons that they held such attitudes; nurses reported avoiding contact with and holding stereotypes about lesbian and gay individuals, but they did not report high levels of holding judgmental attitudes toward lesbian and gay individuals (e.g., perceiving them as immoral, condemning their orientation) [22]. As attitudes may change over time, the trend of changes in attitudes toward homosexuality among Taiwanese nurses must be regularly examined. According to the findings of the World Values Survey, Taiwan made the greatest progress in acceptance of lesbian and gay individuals between 1995 and 2012 compared with China, Japan, and South Korea [23]. Comparing nurses’ attitudes toward homosexuality in 2017 by using the same measures as those used in 2005, and comparing the results obtained in 2017 with those obtained in 2005, may provide directions for developing further educational programs to enhance nurses’ ability to provide nursing care to lesbian and gay patients.

A review study demonstrated that older age, being married, being highly religious, having a low education level, and lacking personal or professional contact with lesbian and gay individuals predicted a negative attitude toward homosexuality among nurses [24]. A recent study reported that nurses who self-identified as heterosexual had poor knowledge regarding homosexuality [25]. Women were also found to have more positive attitudes toward homosexuality compared with men [26], whereas there was no difference in non-acceptance of lesbian and gay patients between female and male nursing staff and students [27]. Whether sociodemographic characteristics are associated with the changing trend in attitudes toward homosexuality among nurses over a period of 12 years in Taiwan should be investigated.

### 1.6. Study Aims and Hypotheses

The present study was conducted in Taiwan using the same instrument as that used in 2005 to examine the levels of various dimensions of attitudes toward lesbian and gay individuals among nurses in 2017. The first aim of this study was to compare attitudes observed among nurses in 2005 with those observed in 2017. The second aim was to examine the roles of demographic characteristics in the changing trend of attitudes toward homosexuality between 2005 and 2017. The third aim was to examine the sociodemographic factors of attitudes toward homosexuality among nurses in 2005 and in 2017 separately. We developed three hypotheses. First, compared with nurses in 2005, nurses in 2017 had more favorable attitudes toward homosexuality. Second, we hypothesized that sociodemographic characteristics moderated the changes in attitudes toward homosexuality in nurses from 2005 to 2017. Third, the relationships of sociodemographics with attitudes toward homosexuality among nurses may have varied between Survey 2005 and Survey 2017.

## 2. Methods

### 2.1. Participants

We recruited participants from a medical center and two regional hospitals in Southern Taiwan using the same processes in Survey 2005 and Survey 2017. We invited each nurse to anonymously complete questionnaires and deliver them to us during July to August 2005 (Survey 2005) and during October to December 2017 (Survey 2017.). In total, 1176 nurses (a response rate of 84.4%) in Survey 2005 and 1519 nurses (a response rate of 73.4%) were included in this study and returned the questionnaire without any omission. In Survey 2005, 97.9% and 96.5% of respondents who provided incomplete data were female in 2005 and in 2017, respectively; 53.3% and 62.9% of respondents who provided incomplete data were older than 30 in 2005 and in 2017, respectively. Compared with the age and sex of participants whose data were used in the final analysis (Table 1), no significant difference in sex and age existed between respondents who did and did not provide incomplete data in 2005 and 2017 (all *p* > 0.05). The Institutional Review Board of Kaohsiung Medical University approved the study protocol (approval number: KMUHIRB-F(I)-20160065 and date of approval: 20 December 2016).

### 2.2. Survey Instruments

#### 2.2.1. Attitudes toward Homosexuality

We used the Taiwanese version of the Attitudes Toward Homosexuality Questionnaire (ATHQ) to measure the level of nurses’ attitudes toward homosexuality [28]. The ATHQ contained 49 questions and examined four dimensions of attitudes toward homosexuality: condemnation (11 items; e.g., “Lesbians (gay men) should be required to register with the police department where they live.”), immorality (13 items; e.g., “Gay men (lesbians) endanger the institution of the family.”), avoiding contact (18 items; e.g., “I avoid gay men (lesbians) whenever possible.”), and stereotypes (7 items; e.g., “The love between two lesbians (gay men) is quite different from the love between two persons of the opposite sex.”). Participants specified their level of agreement with a statement according to five points: (1) strongly disagree; (2) disagree; (3) neither agree nor disagree; (4) agree; (5) strongly agree. A higher score indicates a more negative attitude toward homosexuality. Assessment of Survey 2005 revealed that the Taiwanese version of the ATHQ had acceptable reliability and validity [22].

#### 2.2.2. Sociodemographic Characteristics

Participants’ sex, age, education level (below Bachelor’s degree vs. having a Bachelor’s or above), frequency of attending religious activities (not regular vs. regular), and self-identified sexual orientation were examined. The participants’ age distribution was highly skewed; thus, participants were further divided into two groups according to age (aged ≤30 years vs. aged >30 years). We also examined participants’ tenure as nurses. Participants’ tenure was highly correlated with age (*r* of Pearson’s correlation = 0.940); thus, tenure as a nurse was not considered in the analysis in the present study.

### 2.3. Statistical Analysis

The sociodemographic characteristics and attitudes of participants in Survey 2005 and those in Survey 2017 toward lesbian and gay individuals were compared by using χ^2^ and *t* tests. The levels of attitudes toward homosexuality in the four dimensions of the ATHQ between nurses in Survey 2005 and those in Survey 2017 were compared by performing multiple regression analysis to control for sociodemographic characteristics. The moderating effects of sociodemographic characteristics on the association between survey year and attitudes toward homosexuality were also examined. Because of multiple comparisons for the four dimensions of the ATHQ, a two-tailed *p* value of less than 0.0125 (0.05/4) was considered statistically significant.

## 3. Results

### 3.1. Comparison of Attitudes toward Homosexuality between Participants in Surveys 2005 and 2017

Participants in Survey 2017 were older than those in Survey 2005. The proportions of participants in Survey 2017 who were male, had a Bachelor’s or above, and attended religious activities regularly were higher than those in Survey 2005. Table 1 also lists the attitudes toward homosexuality of participants in Surveys 2005 and 2017. Participants in Survey 2017 exhibited higher levels of condemnation of gay and lesbian individuals and a higher tendency to perceive gay and lesbian individuals as immoral than did those in Survey 2005, whereas participants in Survey 2017 exhibited lower levels of avoiding contact with and holding stereotypes toward lesbian and gay individuals than did those in Survey 2005.

The effect of survey year on nurses’ attitudes toward homosexuality was examined by performing multiple regression analysis (Model I in Table 2). After controlling for the effects of demographic characteristics, participants in Survey 2017 exhibited higher levels of condemnation of homosexuality and perceptions of it as immoral than did those in Survey 2005, whereas participants in Survey 2017 exhibited lower levels of avoiding contact with lesbian and gay individuals and stereotypes toward homosexuality than did those in Survey 2005.

### 3.2. Sociodemographic Moderators of Changes in Attitudes toward Homosexuality

We examined the moderating effects of demographics on the association between survey year and attitudes toward homosexuality (Mode II in Table 2). The results indicated that the interaction between survey year and age group was significantly associated with condemnation, avoiding contact, and stereotypes, indicating that the age group moderated the associations of survey year with condemnation, avoiding contact, and stereotypes. The results of further analysis revealed that deepened condemnation toward homosexuality from 2005 to 2017 was more significant in nurses older than 30 years (β = 0.427, *p* < 0.001) than in nurses aged 30 years or younger (β = 0.219, *p* < 0.001). The reduction in avoiding contact with lesbian and gay individuals from 2005 to 2017 was more significant in nurses aged 30 years or younger (β = −0.306, *p* < 0.001) than in nurses older than 30 years (β = −0.151, *p* < 0.001). A decreased level of stereotypes toward homosexuality from 2005 to 2017 was more significant in the nurses aged 30 years or younger (β = −0.453, *p* < 0.001) than in nurses older than 30 years (β = −0.103, *p* = 0.007).

### 3.3. Factors Related to Attitudes toward Homosexuality in Survey 2005 and Survey 2017

We further examined the relationships of age, sex, education level, and frequency of attending religious activities with attitudes toward homosexuality in Survey 2005 and Survey 2017 separately (Table 3). The results revealed that compared with nurses aged 30 years or younger, nurses older than 30 years exhibited more negative attitudes of all dimensions in both Survey 2005 and 2017, except for stereotypes in Survey 2005. Female nurses had higher levels of condemnation and stereotypes than male nurses in Survey 2005 but not in Survey 2017. Nurses with a Bachelor’s or Master’s degree had lower condemnation and avoidant contact in both Survey 2005 and 2017 than those without a Bachelor’s or Master’s degree, whereas the associations of education level with immorality and stereotypes existed only in Survey 2005 but not in Survey 2017. Nurse who regularly attended religious activities had higher immorality than those who did not regularly attend religious activities, whereas the association of regularly attending religious activities with avoiding contact existed only in Survey 2017 but not in Survey 2005.

## 4. Discussion

The present study demonstrated that nurses in 2017 exhibited lower levels of avoiding contact with lesbian and gay individuals and stereotypes toward homosexuality but higher levels of condemnation of gay and lesbian individuals and perceptions of gay and lesbian individuals as immoral than did nurses in 2005. Age and sexual orientation moderated changes in some dimensions of attitudes toward homosexuality from 2005 to 2017.

### 4.1. Various Directions of Changes across Four Dimensions of Attitudes toward Homosexuality

Given that the World Values Survey found that the level of tolerance toward homosexuality in Taiwanese people increased by 132% from 1995 to 2012, which is considerably higher than that for Japan (64%), South Korea (60%), and China (58%) [23], it would be reasonable to suspect that the acceptance of homosexuality among Taiwanese nurses would have also increased notably from 2005 to 2017. However, the present study showed that the direction of change in attitudes varied in terms of the various dimensions of attitudes toward homosexuality. First, the levels of avoiding contact with lesbian and gay individuals and stereotypes toward homosexuality decreased from 2005 to 2017. Lesbian and gay individuals in Taiwan may feel safer and may be comfortable in revealing their sexual orientation to others because of the change in social attitude. Meanwhile, the popularity of the Internet has increased opportunities for people to learn new things and concepts that diverge from traditional ones, including those regarding the diversity of sexual orientation [29]. Therefore, Taiwanese nurses in 2017 presumably had more opportunities to interact with lesbian and gay individuals in their daily lives and nursing practice than did those in 2005; this was reflected in nurses in 2017 exhibiting lower levels of avoiding contact with and having stereotypes about lesbian and gay individuals.

However, the levels of condemnation of homosexuality and perceptions of gay and lesbian individuals as immoral among nurses in Taiwan did not decrease but rather became more intense from 2005 to 2017. Both condemnation of homosexuality and perceiving gay and lesbian individuals as immoral are value judgments derived from heterosexism. The present study could not determine why changes in directions varied among the various dimensions of attitudes toward homosexuality among Taiwanese nurses. The last report of the World Values Survey regarding Taiwanese people’s attitudes toward homosexuality was conducted in 2012. Therefore, additional studies should be conducted to examine whether changes have occurred in social attitudes toward homosexuality in the general population in Taiwan since 2012. Meanwhile, in Taiwan, there are no curricula or training courses developed and provided to nursing students and nurses to enhance their professional competence in caring for sexual minorities [25]. The present study indicates that such training courses are urgently required.

### 4.2. Roles of Age

The findings of the present study demonstrate that the attitude of condemnation deepened more and that of avoiding contact with and having stereotypes toward lesbian and gay individuals increased from 2005 to 2017 in nurses older than 30 years compared with nurses aged 30 years or younger. Younger nurses may have a stronger ability to consider and accept concepts that are different from traditional values, such as those pertaining to the diversity of sexual orientation. The results of this study indicated that professionals should take age into consideration when developing training programs for nurses to provide culturally competent care to sexual minorities. Older nurses may encounter challenges in not only learning new concepts of nursing care but also their belief toward “normal sexual orientation.” However, this study found that the attitude of condemnation toward homosexuality deepened even in nurses aged 30 years or younger from 2005 to 2017. These findings indicate that training programs are necessary for nurses of all age groups.

### 4.3. Sociodemographics Related to Attitudes toward Homosexuality

The present study demonstrated that several sociodemographics were significantly associated with the levels of attitudes toward homosexuality in both Survey 2005 and Survey 2017, whereas some significant associations were found in Survey 2005 or Survey 2017. For example, the significant associations of lower education level with immorality and stereotypes existed only in Survey 2005 but not in Survey 2017; being female was significantly associated with higher levels of condemnation and stereotypes in Survey 2005 but not in Survey 2017. In addition to the differences in the proportions of participants regarding sex and education level between Survey 2005 and Survey 2017, other factors, such as the change in social attitudes toward homosexuality in the general population, may also contribute to the various effects of sex and education level on attitudes toward homosexuality between Survey 2005 and Survey 2017. It is worth noting that the significant association of regularly attending religious activities with avoiding contact existed only in Survey 2017 but not in Survey 2005. The levels of tolerance toward homosexuality in believers may vary by religion, year, and region. For example, the World Values Survey in Taiwan demonstrated that Christians were not especially intolerant toward homosexuality in 1995 but became significantly less tolerant than other religions by 2012, which is likely due to a resistance to attitudinal changes in the Christian community [23].

### 4.4. Limitations

The present study did not examine factors that may explain the changes in attitudes toward homosexuality among nurses in Taiwan over the period between 2005 and 2017.

## 5. Conclusions

Some dimensions of negative attitudes toward lesbian and gay individuals decreased but some dimensions deepened among nurses between 2005 and 2017. The need to develop training programs aimed at improving not only the quality of nursing skills but also their negative attitudes (condemnation, perceived immorality) regarding homosexuality is urgent. When developing such training programs, professionals should take the age of nurses into account to achieve the best results.

## Figures and Tables

**Table 1 ijerph-18-03465-t001:** Sociodemographic characteristics and attitudes toward homosexuality among nurses in Survey 2005 and Survey 2017.

Variable	Survey 2005 (N = 1176)		Survey 2017 (N = 1519)		*t* or χ^2^	*p*
	Mean (SD)	%	Mean (SD)	%		
Sociodemographic characteristics						
Age (year)	29.2 (6.2)		34.7 (8.3)		−18.945	<0.001
Age group						
30 or younger		68.7		36.9	268.871	<0.001
Older than 30		31.3		63.1		
Sex						
Male		0.9		2.3	8.533	0.003
Female		99.1		97.7		
Education level						
below Bachelor’s degree		62.1		8.3	884.521	<0.001
Bachelor’s or Master’s degree		37.9		91.7		
Attendance of religious activities						
No or irregular		48.2		42.8	7.871	0.005
Regular		51.8		57.2		
Duration of being a nurse (year)	6.8 (5.9)		11.9 (8.3)		−17.898	<0.001
Attitudes toward homosexuality on the ATHQ						
Condemnation	2.3 (0.5)		2.7 (0.4)		−19.215	<0.001
Immorality	2.5 (0.5)		2.7 (0.4)		−8.580	<0.001
Avoiding contact	2.8 (0.5)		2.5 (0.3)		13.662	<0.001
Stereotypes	2.8 (0.5)		2.5 (0.6)		14.107	<0.001

ATHQ: Attitudes Toward Homosexuality Questionnaire.

**Table 2 ijerph-18-03465-t002:** Factors related to attitudes toward homosexuality: multiple regression analysis.

Variables	Condemnation	Immorality	Avoiding Contact	Stereotypes
	B (SE)	B (95%CI)	B (95%CI)	B (95%CI)
Model 1				
Survey 2017 (reference = Survey 2005)	0.329 (0.021) *	0.159 (0.021) *	−0.224 (0.020) *	−0.278 (0.027) *
Older than 30 (reference = 30 or younger)	0.193 (0.017) *	0.180 (0.018) *	0.166 (0.017) *	0.157 (0.022) *
Females (reference = males)	0.152 (0.063)	0.142 (0.066)	0.025 (0.063)	0.251 (0.082) *
Bachelor’s or Master’s degree (reference = below Bachelor’s degree)	−0.126 (0.021) *	−0.127 (0.022) *	−0.108 (0.021) *	−0.139 (0.028) *
Regular attendance of religious activities (reference = no or irregular attendance)	0.055 (0.016) *	0.065 (0.017) *	0.045 (0.016) *	0.043 (0.021)
Model 2 ^a^				
Survey year × age group	0.131 (0.035) *	0.034 (0.036)	0.110 (0.035) *	0.321 (0.045) *
Survey year × sex	−0.315 (0.152)	−0.330 (0.157)	−0.370 (0.151)	−0.278 (0.196)
Survey year × education level	−0.018 (0.047)	0.064 (0.049)	0.023 (0.047)	0.022 (0.061)
Survey year × frequency of attending religious activities	−0.014 (0.033)	−0.033 (0.034)	0.027 (0.033)	−0.059 (0.042)

^a^ Controlling for survey year, age group, sex, education level, and frequency of attending religious activities. CI: confidence interval; SE: standard error; * *p* < 0.0125.

**Table 3 ijerph-18-03465-t003:** Factors related to attitudes toward homosexuality in Survey 2005 and Survey 2017: multiple regression analysis ^a^.

Variables	Condemnation		Immorality		Avoiding Contact		Stereotypes	
	B (SE)		B (SE)		B (SE)		B (SE)	
	Survey 2005	Survey 2017	Survey 2005	Survey 2017	Survey 2005	Survey 2017	Survey 2005	Survey 2017
Older than 30 (reference = 30 or younger)	0.116 (0.029) *	0.247 (0.021) *	0.165 (0.034) *	0.199 (0.018) *	0.102 (0.033) *	0.212 (0.017) *	−0.026 (0.029)	0.295 (0.032) *
Females (reference = males)	0.396 (0.145) *	0.082 (0.066)	0.397 (0.171)	0.067 (0.058)	0.313 (0.167)	−0.057 (0.053)	0.465 (0.148) *	0.187 (0.102)
Bachelor’s or Master’s degree (reference = below Bachelor’s degree)	−0.107 (0.028) *	−0.125 (0.036) *	−0.140 (0.033) *	−0.076 (0.032)	−0.102 (0.032) *	−0.079 (0.029) *	−0.116 (0.028) *	−0.094 (0.056)
Regular attendance of religious activities (reference = no or irregular attendance)	0.058 (0.027)	0.044 (0.020)	0.082 (0.031) *	0.049 (0.018) *	0.026 (0.031)	0.053 (0.016) *	0.065 (0.027)	0.006 (0.031)

^a^ Controlling for survey year, age group, sex, education level, and frequency of attending religious activities. CI: confidence interval; SE: standard error; *: *p* < 0.0125.

## Data Availability

Restrictions apply to the availability of these data. Only researchers of this study can access the data.

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
