# Peer review of "Attitudes toward Homosexuality among Nurses in Taiwan: Effects of Survey Year and Sociodemographic Characteristics"

_ijerph, 2021, doi:10.3390/ijerph18073465_

Round 1
Reviewer 1 Report
The authors describe attitudes towards homosexuality in Taiwanese nurses over time. The population is highly informative, the analysis is well contextualized and competently carried out.
The purpose of some analyses remains obscure, however, and I would suggest a more direct analytic (and simpler) approach to maximize the insight to be gained from this exceptional dataset.
Specifically, the analysis in Table 1 demonstrates considerable change in attitudes over this period, and in opposite directions for subscales of the ATHQ, a curious finding indeed!
However, the analysis in Table 2 merely includes year of survey as a covariate, and expends considerable ink on interactions terms that are nearly impossible to interpret directly (see Rothman, Greenland, Lash, Chapter 5 "Concepts of Interaction"). A more informative analysis would be to model the correlates of attitudes in 2005 and of those in 2017 separately, and comparing these findings to elucidate the degree to which the structure of heteronormative views has changed over time. In short, model 2 offers no insight on the questions that Table 1 raises.
A small point, the group labeled as "not absolutely heterosexual" appears to be quite heterogeneous, making up 100 of the 101 points on the 0-100 scale. This reviewer suspects that claiming a position of "100" rather than a slightly lower value may be more a statement of attitudes itself, rather than a claim about sexual orientation identity. If so, adjusting for this variable as currently dichotomized is probably an overadjustment for a strong correlate of attitude, rather than identity. Alternate strategies would include dropping the variable entirely from the analysis, or dichotomizing it at a point more likely to represent a break between heterosexual and sexual minority respondents (just guessing here, but maybe 67 or 76 or something like that).
Presentation of findings:
2.1 Participants - rather than use a p-value cut-off, provide actual percents of these characteristics (sex, hospital type, age) for respondents and non-respondents.
Table 1 - percents only, no need for n's as well.
The p-values in table 2 are distracting and redundant to the t values presented. The table would be much easier to read with betas and standard errors (or betas and confidence intervals) than t and p values.
Author Response
Comment 1
The purpose of some analyses remains obscure, however, and I would suggest a more direct analytic (and simpler) approach to maximize the insight to be gained from this exceptional dataset. Specifically, the analysis in Table 1 demonstrates considerable change in attitudes over this period, and in opposite directions for subscales of the ATHQ, a curious finding indeed! However, the analysis in Table 2 merely includes year of survey as a covariate, and expends considerable ink on interactions terms that are nearly impossible to interpret directly (see Rothman, Greenland, Lash, Chapter 5 "Concepts of Interaction"). A more informative analysis would be to model the correlates of attitudes in 2005 and of those in 2017 separately, and comparing these findings to elucidate the degree to which the structure of heteronormative views has changed over time. In short, model 2 offers no insight on the questions that Table 1 raises.
Response
Thank you for your comment. In the revised manuscript, examining the correlates of attitudes in 2005 and of those in 2017 was listed as the third aim of this study. We reanalyzed the data and examined the sociodemographic correlates of attitudes in 2005 and in 2017 separately. We did find that the sociodemographic correlates of attitudes were not exactly the same. We added the new results and discussion into the revised manuscript as below.
“We also examined the sociodemographic factors of attitudes toward homosexuality among nurses in 2005 and in 2017 separately.”
“Third, the relationships of sociodemographics with attitudes toward homosexuality among nurses may vary between Survey 2005 and Survey 2017.”
“3.3. Factors Related to Attitudes Toward Homosexuality in Survey 2005 and Survey 2017
We further examined the relationships of age, sex, education level and frequency of attending religious activities with attitudes toward homosexuality in Survey 2005 and Survey 2017 separately (Table 3). The results revealed that compared with nurses aged 30 years or younger, nurses older than 30 years exhibited more negative attitudes of all dimensions in both Survey 2005 and 2017 except for stereotypes in Survey 2005. Female nurses had higher levels of condemnation and stereotypes than male nurses in Survey 2005 but not in Survey 2017. Nurses with a bachelor's or master's degree had lower condemnation and avoidant contact in both Survey 2005 and 2017 than those without bachelor's or master's degree, whereas the associations of education level with immorality and stereotypes existed only in Survey 2005 but not in Survey 2017. Nurse who regularly attended religious activities had higher immorality than those who did not regularly attend religious activities, whereas the association of regularly attending religious activities with avoiding contact existed only in Survey 2017 but not in Survey 2005.”
“4.3. Sociodemographics Related to Attitudes Toward Homosexuality
The present study demonstrated that several sociodemographics were significantly associated with the levels of attitudes toward homosexuality in both Survey 2005 and Survey 2017, whereas some significant associations were found in Survey 2005 or Survey 2017. For example, the significant associations of lower education level with immorality and stereotypes existed only in Survey 2005 but not in Survey 2017; being female was significantly associated with higher levels of condemnation and stereotypes in Survey 2005 but not in Survey 2017. In addition to the differences in the proportions of participants regarding sex and educaiton level between Surevy 2005 and Survey 2017, other factors such as change in social attitudes toward homosexuality in the general population may also contribute to the various effects of sex and educaiton level on attitudes toward homosexuality between Survey 2005 and Survey 2017. It is worth noting that the significant association of of regularly attending religious activities with avoiding contact existed only in Survey 2017 but not in Survey 2005. The levels of tolerance toward homosexuality in believers may vary by religions, years and regions. For example, the World Values Survey in Taiwan demonstrated that Christians were not especially intolerant toward homosexuality in 1995, but became significantly less tolerant than other religions by 2012, which is likely due to a resistance to attitudinal changes for the Christian community [23].
Comment 2
A small point, the group labeled as "not absolutely heterosexual" appears to be quite heterogeneous, making up 100 of the 101 points on the 0-100 scale. This reviewer suspects that claiming a position of "100" rather than a slightly lower value may be more a statement of attitudes itself, rather than a claim about sexual orientation identity. If so, adjusting for this variable as currently dichotomized is probably an overadjustment for a strong correlate of attitude, rather than identity. Alternate strategies would include dropping the variable entirely from the analysis, or dichotomizing it at a point more likely to represent a break between heterosexual and sexual minority respondents (just guessing here, but maybe 67 or 76 or something like that).
Response
Thank you for your suggestion. We dropped the variable “absolutely heterosexual or not” entirely from the analysis. The results of reanalysis are the same as those of original analysis. Please refer to Results section.
Comment 3
2.1 Participants - rather than use a p-value cut-off, provide actual percents of these characteristics (sex, hospital type, age) for respondents and non-respondents.
Response
Thank you for your suggestion. We added the actual percentages of sex and age into the revised manuscript as below.
“In Survey 2005, 97.9% and 96.5% of respondents who provided incomplete data were female in 2005 and in 2017, respectively; 53.3% and 62.9% of respondents who provided incomplete data were older than 30 in 2005 and in 2017, respectively. Compared with the age and sex of participants whose data were used in the final analysis (Table 1), no significant difference in sex and age existed between respondents who did and did not provide incomplete data in 2005 and 2017 (all p > 0.05).”
Comment 4
Table 1 - percents only, no need for n's as well.
Response
We removed the numbers and left percents in Table 1.
Comment 5
The p-values in table 2 are distracting and redundant to the t values presented. The table would be much easier to read with betas and standard errors (or betas and confidence intervals) than t and p values.
Response
We changed the presentation in Table 2 by replacing t and p by betas and standard errors.
Reviewer 2 Report
It is a relevant theme. The article is reasonably well written, consider reviewing the various sentences that begin "In Addition". Reviewing references in the text page 2 [0,6-8] ,[00],[03].
Consider reviewing the 1.5 study aims about definitions of hypotheses.
In the methodology, consider including the type of study and making the relationship between the type of study and the definition of the 2 hypotheses that you defined. Consider reviewing “No difference in sex (p > 0.05), type of hospital (p > 0.05), or age (p > 0.05) was found between respondents who did and did not provide incomplete data in 2005 and
2017.” It is not necessary to repeat the p-value 3 times in the same sentence.
Consider reviewing the study limitations. The second limitation that you describe is not a limitation of the study.
Author Response
Comment 1
The article is reasonably well written, consider reviewing the various sentences that begin "In Addition".
Response
Thank you for your comment. We reviewed the six sentences beginning with “In addition” (four in Introduction section and two in Methods section) and revised them in the revised manuscript.
Comment 2
Reviewing references in the text page 2 [0,6-8] ,[00],[03].
Response
Thank you for your reminding. They might be produced by the process of changing the format for review. We reviewed and revised them.
Comment 3
Consider reviewing the 1.5 study aims about definitions of hypotheses. In the methodology, consider including the type of study and making the relationship between the type of study and the definition of the 2 hypotheses that you defined.
Response
Thank you for your comment. We revised the hypotheses of this study as below.
“The third aim was to examine the sociodemographic factors of attitudes toward homosexuality among nurses in 2005 and in 2017 separately. We have three hypotheses. First, compared with nurses in 2005, nurses in 2017 have more favorable attitudes toward homosexuality. Second, we hypothesized that sociodemographic characteristics moderate the changes of attitudes toward homosexuality in nurses from 2005 to 2017. Third, the relationships of sociodemographics with attitudes toward homosexuality among nurses may vary between Survey 2005 and Survey 2017.”
Comment 4
Consider reviewing “No difference in sex (p > 0.05), type of hospital (p > 0.05), or age (p > 0.05) was found between respondents who did and did not provide incomplete data in 2005 and 2017.” It is not necessary to repeat the p-value 3 times in the same sentence.
Response
Thank you for your comment. We revised the sentence as below.
“In Survey 2005, 97.9% and 96.5% of respondents who provided incomplete data were female in 2005 and in 2017, respectively; 53.3% and 62.9% of respondents who provided incomplete data were older than 30 in 2005 and in 2017, respectively. Compared with the age and sex of participants whose data were used in the final analysis (Table 1), no significant difference in sex and age existed between respondents who did and did not provide incomplete data in 2005 and 2017 (all p > 0.05).”
Comment 5
Consider reviewing the study limitations. The second limitation that you describe is not a limitation of the study.
Response
Thank you for your comment. We removed the second limitation from the revised manuscript.
Reviewer 3 Report
Thank you for this well written, important paper. I was very intrigued by your findings! My only real feedback is this:
I would recommend using some reports/data from your area for the background section. It's great to highlight the IOM and HealthyPeople 2020, but it would also be good to highlight efforts like these in Taiwan. I know there is a broad audience for this journal, but I also think it's important to highlight efforts like these in your area of the world.
Author Response
Comment
I would recommend using some reports/data from your area for the background section. It's great to highlight the IOM and HealthyPeople 2020, but it would also be good to highlight efforts like these in Taiwan. I know there is a broad audience for this journal, but I also think it's important to highlight efforts like these in your area of the world.
Response
Thank you for your comment. We added the efforts of improving gender equality in Taiwan as below.
1.2. Efforts to Integrate Gender into Nursing and Medical Education
Taiwan has made significant progresses in gender equality in the past two decades. Taiwan ratified the Convention on the Elimination of all Forms of Discrimination Against Women (CEDAW) in 2007 to elevate the standard of gender rights in the country and advance gender equality; the government also promulgated the Enforcement Act of CEDAW in 2012 to make CEDAW provisions effective as domestic law [5]. The government also promulgated the Gender Equality Policy Guidelines in 2011 to establish the directions of administrative politics of gender equality [6]. Moreover, the government formulated the Gender Equality Education Act in 2015 to eliminate gender discrimination, safeguard human dignity, and establish education resources and environments that epitomize gender equality [7]. The rights of sexual minority individuals also got attention by the society.
Regarding health care, the White Paper on Taiwanese Medical Education in 2002 firstly revealed the importance of integrating the concept of gender equality into medical education [8]. The Ministry of Health and Welfare in Taiwan amended the Healthcare Professionals Registration and Continuing Education Guidelines in 2016 to incorporate gender topics as the compulsory credits for all healthcare professionals [9]. However, compared with the health needs of women, the health need of sexual minority individuals have less been emphasized in Taiwan. Given that nurses are important healthcare providers, several professionals called for enhancing nurses’ competencies of healthcare for sexual minority individuals [10-13].
- Executive Yuan, Taiwan. Protection of Women's Rights Under CEDAW. Available online: https://english.ey.gov.tw/News3/9E5540D592A5FECD/f7816790-ad30-49fe-88e9-763a7c9e7f9f (accessed on 16 March 2021).
- Executive Yuan, Taiwan. Gender Equality Policy Guidelines. Available online: https://gec.ey.gov.tw/Page/FD420B6572C922EA (accessed on 16 March 2021).
- Ministry of Education, Taiwan. Gender Equality Education Act. Available online: https://law.moj.gov.tw/ENG/LawClass/LawAll.aspx?pcode=H0080068 (accessed on 16 March 2021).
- Huang, K.Y. The White Paper on Taiwanese Medical Education. Medical Education Committee of the Ministry Education, Taipei; 2002.
- Yang, H.C.; Yen, C.F. Integrating gender into medicine: Research on the construction of gender competence indicators in medical education. Taiwan J Sociol Educ. 2018, 18, 91-145. doi: 10.3966/168020042018061801003.
- Hsu, C.Y. Nursing competencies of healthcare for sexual/gender minorities. National Taiwan University Hospital Journal of Nursing (Taiwan). 2018, 14, 6-17. doi: 10.6740.NTUHJN.201801_14(1).0003.
- Lee, Y.W. Coming out of homosexuality and its nursing care. Chang Gung Journal of Science, 2019, 30, 87-95. doi: 10.6192/CGUST.201906_(30).8.
- Tian, S.H.; Shen, M.H. The implementation of the right to equality in the care of a vulnerable medical population: Pregnant and postpartum lesbians. The Journal of Nursing (Taiwan), 2019, 66, 65-71. doi: 10.6224/JN.201910_66(5).09.
- Wei, H.T.; Chen, M.H.; Ku, W.W. Fostering LGBT-friendly healthcare services. The Journal of Nursing (Taiwan), 2015, 62, 22-28. doi: 10.6224/JN.62.1.22.